# STOCHASTOK: IMPROVING FINE-GRAINED SUBWORD UNDERSTANDING IN LLMS

**Anya Sims**[1†]    **Klara Kaleb**[1]    **Jakob Nicolaus Foerster**[1]

**Yee Whye Teh**[1]    **Cong Lu**[2]

[1]University of Oxford   [2]University of British Columbia

## ABSTRACT

Despite impressive performance, large language models (LLMs) still struggle with seemingly simple questions such as "How many r's are in 'strawberry'?" This limitation highlights that LLMs are unable to understand how humans 'see' language. We attempt to address this by experimenting with stochastic tokenization schemes in which the same text may be tokenized into multiple possible token sequences. We find that using stochastic tokenization during pretraining dramatically alters the representations learned and allows LLMs to capture understanding of fine-grained spelling-level detail in addition to the structure learned with standard tokenization. We demonstrate this by showing that LLMs pretrained with standard deterministic tokenization cannot be finetuned to answer language-game type questions, whilst with the minimal addition of stochastic tokenization during pretraining, the corresponding LLMs perform near-perfectly. Crucially, these improvements are achieved without any performance drop on standard benchmarks or any additional training cost — the only change is a single simple, computationally cheap preprocessing step. Overall, our results suggest that embracing stochastic tokenization can help enable LLMs to better understand how humans perceive language.

## 1 INTRODUCTION

Large language models (LLMs) have achieved remarkable progress on a wide range of tasks (Achiam et al., 2023; Team et al., 2023; Dubey et al., 2024). However, their reliance on tokenization obscures how humans naturally perceive language. For example, while humans see 'book' and 'cook' as differing by a single letter, LLMs always see these words as distinct token IDs[1]. This makes subword-focused tasks such as counting letters or identifying shared substrings difficult, even for current state-of-the-art LLMs. While these weaknesses may seem limited to wordplay-based games, they highlight a more fundamental inability of LLMs to understand how humans perceive language, an essential aspect of being able to communicate with humans effectively.

In light of this, we investigate whether stochastic tokenization can address these limitations, where 'stochastic tokenization' refers to any tokenization scheme in which the same text may be encoded as multiple possible token sequences. We start with the simplest instantiation of this, in which tokens are randomly split into equivalent pairs of smaller tokens with some small probability. Our experiments show that this minimal preprocessing step significantly alters the representations learned and allows the model to capture human notions of written language in addition to the structure learned with deterministic tokenization. We demonstrate this by showing that language models pretrained with stochastic tokenization can quickly adapt to near-perfect accuracy on 'language game' tasks, while the models trained with deterministic tokenization fail. Crucially, the benefit comes without any performance drop on the original benchmarks or any additional training computation cost.

It is also notable that stark performance change is achieved with the simplest instantiation of stochastic tokenization, which comes with several benefits. Firstly, while prior methods such as subword regularization (Kudo, 2018) and BPE-dropout (Provilkov et al., 2020) rely on Unigram and BPE

---

[†]Corresponding author `anya.sims@stats.ox.ac.uk`
[1]e.g. 'book'=`3092` and 'cook'=`171691` in the GPT-4o and GPT-4o mini models.

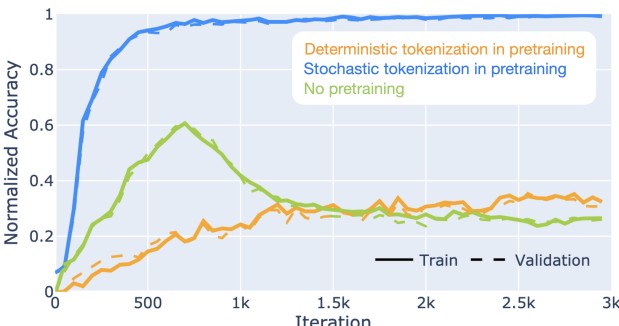

Figure 1: **Main result**: Tokenizing the pretraining dataset stochastically means the representations learned during pretraining capture the fine-grained details of how humans 'see' language. This is demonstrated as models pretrained with stochastic tokenization can be finetuned to answer language game questions.

tokenization respectively, our method is compatible with any base tokenizer. Furthermore, unlike prior methods, our algorithm retains the same vocabulary and decoding function as the original tokenizer, making it simple to switch between stochastic and standard tokenization in different stages of model training[2]. Finally, it has minimal computational overhead, while subword regularization requires using the Viterbi algorithm (Viterbi, 1967), or Forward-Filtering and Backward-Sampling (FFBS, Scott (2002)), and BPE-dropout requires retokenizing the data from scratch for different stochasticity levels.

Tokenization has recently received less attention than finetuning and other downstream tasks since its position at the start of the pretraining pipeline often means experimentation is prohibitively expensive. However, the striking difference in behavior achieved with such a modest change may highlight additional benefits—including improved handling of typos, enhanced robustness to training data quality, reduced susceptibility to overfitting due to increased randomness, as well as potentially helping solve the 'counting' problem described by Zhang et al. (2024); Barbero et al. (2024). Overall, our findings suggest that stochastic tokenization is a promising direction to revisit.

## 2  BACKGROUND AND RELATED WORK

The two dominant tokenization methods are Byte-Pair Encoding (BPE, Sennrich et al. (2016)) and Unigram tokenization (see Appendix A for a brief summary).[3] The main stochastic variant of BPE is BPE-dropout (Provilkov et al., 2020). In this algorithm, stochasticity is introduced by randomly omitting some of BPE's constructed merge operations during tokenization. This, however, results in a different vocabulary than that of the original BPE and forces tokenization from scratch, which complicates the reuse of pretrained resources and consistency between training and inference.

The main stochastic variant of Unigram is subword regularization (Kudo, 2018). Here, stochasticity is added by sampling valid segmentations from the learned probability distribution rather than choosing the maximum probability segmentation. This, however, adds to the already considerable computational and memory requirements of Unigram due to its reliance on the Viterbi algorithm or forward–backward search (FSBS) and involves many implementation complexities — for example, handling overlapping candidates, tuning beam search parameters, and ensuring numerical stability.

## 3  SIMPLE STOCHASTIC TOKENIZATION

Our proposed stochastic tokenization scheme is straightforward yet effective. First, we tokenize the dataset using a standard deterministic tokenizer. Then, in a post-tokenization step, we apply a random 'expansion' operation in which, for `expand_prop*len(dataset)` iterations, a token is randomly selected and split into an equivalent pair of tokens from within the existing vocabulary. This repeated subword re-segmentation allows the model to observe many alternative decompositions; for example, the training data may contain the word `[example]` as any of: `[[example]]`, `[[exam][ple]]`, `[[ex][ample]]`, `[[ex][am][ple]]`, or `[[e][x][am][ple]]`. A simple illustrative example of the vocabulary and tokenization process is provided in Appendix C.

---

[2]For example here we switch to deterministic tokenization in finetuning experiments.

[3]BPE is currently the most common method due to its simplicity and lower memory requirements compared to Unigram (Groeneveld et al., 2024; Dubey et al., 2024; Team et al., 2024; Jiang et al., 2023; Abdin et al., 2024; Guo et al., 2025; Yang et al., 2024; Biderman et al., 2023).

This approach offers several practical advantages over existing methods such as subword regularization and BPE-dropout. First, it is compatible with any base tokenization method including BPE, WordPiece, or Unigram. Second, rather than running a computationally intensive tokenization procedure multiple times, we tokenize the data only once and then cheaply "detokenize" or expand the dataset for various levels of stochasticity (as controlled by the `expand_prop` parameter; see Algorithm 2). Finally, because the method retains the same vocabulary as the original tokenizer, we can more closely isolate the effect of adding stochasticity, reuse the base tokenizer's detokenization and tokenization functions, and switch between stochastic and deterministic tokenization for the same model—such as during different stages of training. In the next section, we describe several experiments used to evaluate the impact of this stochastic tokenization.

## 4 EXPERIMENTS

We build on the 50M-parameter baseline setup in the open-source SuperTinyLanguageModels repo (Hillier et al., 2024). This setup uses the GPT-2 BPE tokenizer from the `tiktoken`[4] library and pretrains the model on the OpenWebText dataset (Gokaslan & Cohen, 2019). In the following section, we describe results from a series of experiments investigating the effects of stochastic tokenization.

### 4.1 SIGNIFICANT IMPROVEMENTS IN LANGUAGE GAME TASKS

For our first experiment, we set up a dataset of language game questions involving identifying word lengths, suffixes, prefixes, substrings, individual letters, etc. (see Appendix D for examples). We look at the performance of models finetuned on these questions starting from three models: (1) pretrained with normal deterministic tokenization, (2) pretrained with stochastic tokenization (STOCHASTOK), and (3) no pretraining. In Figure 1 we observe that the language model pretrained with stochastic tokenization quickly achieves near-perfect accuracy on the language game questions, while the models pretrained with deterministic tokenization and the model with no pretraining are unable to reach high performance. Notably, all models have the same vocabulary and the same deterministic tokenizer during finetuning, meaning the finetuning stage is identical for all models (the only change is whether the STOCHASTOK 'expansion step' is applied before pretraining). This suggests that stochastically tokenized data results in the model's internal representations capturing fine-grained sub-token-level language understanding in addition to the structure learned with deterministic tokenization.

### 4.2 NO HARM IN ORIGINAL PERFORMANCE

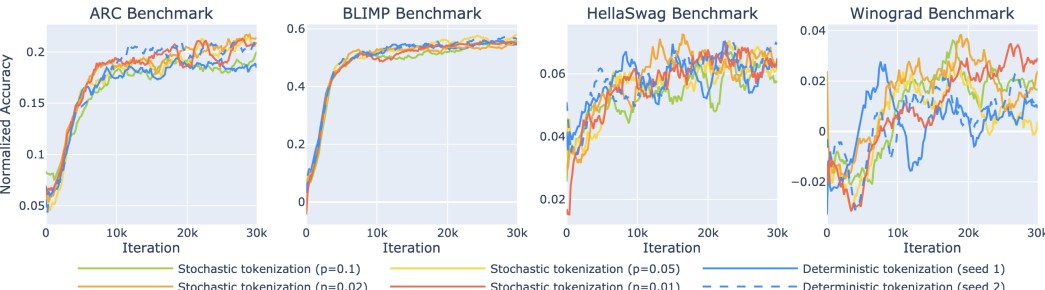

Figure 2: Stochastic tokenization does not harm performance on the original benchmarks.

Crucially, this much-improved fine-grained written language understanding does not come at any cost to the original performance. In Figure 2 we show that stochastic tokenization does not give any performance drop on the original benchmarks (ARC (Clark et al., 2018), Blimp (Warstadt et al., 2020), HellaSwag (Zellers et al., 2019), Winograd Sakaguchi et al. (2021))[5]. Notably, the number of tokens seen during training (and hence computational cost) is fixed for each model, meaning the models trained with stochastic tokenization overall see less text. Furthermore, tokenization stochasticity is added by applying the STOCHASTOK expansion just once before training rather than on-the-fly, meaning the training loop is identical and the only change is a single simple preprocessing step.

---

[4]github.com/openai/tiktoken

[5]We plot normalized accuracy so that 0 is random guessing and 1 is perfect accuracy.

### 4.3 ROBUST TO STOCHASTICITY LEVEL

In Figure 3 we plot the language game accuracy from models pretrained with different stochasticity levels (as controlled by `expand_prop` 'p' in Algorithm 2). We find the benefits of stochastic tokenization to be robust over an order of magnitude range.

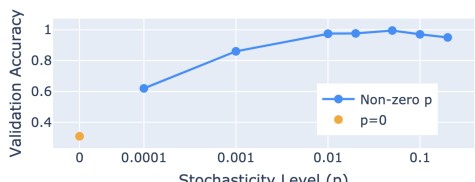

Figure 3: Stochastic tokenization is effective over a wide range of stochasticity levels (log x-scale).

### 4.4 GENERALIZES OUT-OF-DISTRIBUTION

Next, we examine generalization by constructing separate train/validation and holdout language game questions. The train/validation questions all involve identifying substrings/prefixes/suffixes where the substring/prefix/suffix is always less than or equal to half the answer length, while in the holdout set the substring/prefix/suffix is always longer than half the answer length. In Figure 4 we observe that models trained with stochastic tokenization generalize near-perfectly while the deterministic tokenization-pretrained equivalent fails to generalize well.

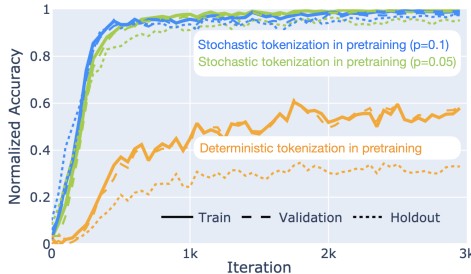

Figure 4: Generalization to heldout language game question types with and without stochastic tokenization pretraining.

### 4.5 INTERNAL REPRESENTATIONS VISIBLY CAPTURE SUBWORD-LEVEL STRUCTURE

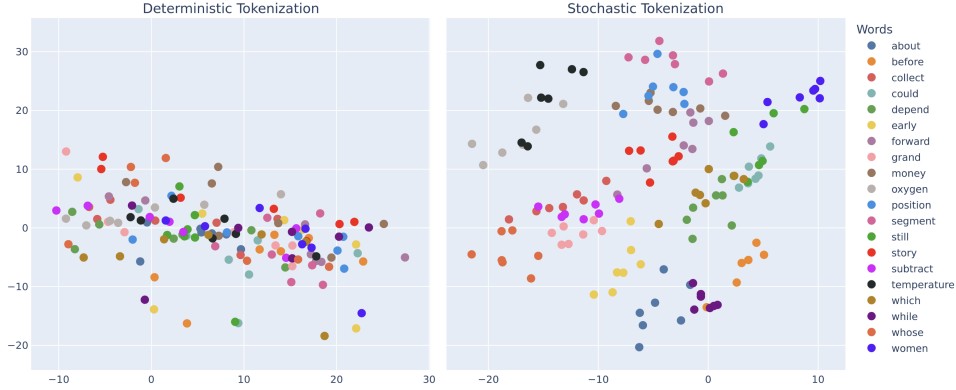

Figure 5: Visualizing the internal representations of models trained with and without stochastic tokenization.

In Figure 5 we visualize the internal representations, both with and without stochastic tokenization. We first fit a PCA model on the embeddings[6] of the top 1k most common words. Next, we plot the PCA-transformed embeddings for alternative tokenizations of the same words, using a random sample of 20 words. We observe that, when using stochastic tokenization, the embeddings for alternative tokenizations of the same word are more closely aligned. This suggests improved morphological awareness and subword-level understanding.

## 5 CONCLUSION

We show that incorporating stochastic tokenization during pretraining dramatically enhances language models' ability to represent subword-level structures central to human language perception. Our experiments demonstrate that a small tweak to tokenization yields dramatic improvements on language game tasks, without compromising—and sometimes even slightly enhancing—standard benchmarks. Recently, research has focused on finetuning and other downstream processes, as experimenting with changes early in the pipeline is often prohibitively computationally expensive. Our results, however, suggest that revisiting tokenization could have a large impact on overall model performance. We are excited by the potential of this approach and hope our work encourages renewed exploration of tokenization schemes to bridge the gap between human and machine language perception.

---

[6]The activations after the final causal attention layer for the last token in the token sequence for each word.

ACKNOWLEDGMENTS

We thank the contributors of OpenWebText and the maintainers of SuperTinyLanguageModels for making their resources publicly available under the MIT License.

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

# SUPPLEMENTARY MATERIAL

## A   A BRIEF SUMMARY OF TOKENIZERS

### A.1   BPE TOKENIZATION

**Construction**
The tokenizer is constructed by initializing the vocabulary as individual characters and then iteratively adding the most frequent adjacent token pair in the "training dataset" until the desired vocabulary size is reached. This yields a vocabulary and a hierarchy of merge rules.

**Encoding**
The dataset is initially tokenized as individual characters. Pairs of tokens are then merged according to the hierarchy of merge rules until there are no more merges available. [7]

**Decoding**
The text strings corresponding to each token ID are simply looked up and joined together.

### A.2   UNIGRAM TOKENIZATION

**Construction**
In contrast to BPE, Unigram starts with a large candidate vocabulary of possible subword units and removes elements to get down to the desired vocabulary size. Tokens are removed from the vocabulary by modeling the dataset as a Unigram model and removing the token that results in the smallest increase in log-likelihood of the dataset considering all possible tokenizations. This relies on using the Viterbi algorithm to compute probabilities of all possible tokenizations. It also relies on using the Expectation-Maximization (EM) to optimize the vocabulary and the probability of the dataset simultaneously. The result is a vocabulary and corresponding probabilities of each token (i.e. a Unigram model of the dataset).

**Encoding**
All possible tokenizations are considered and the one with the highest probability under the unigram model is chosen. This involves using the Viterbi algorithm to find the highest probability tokenization.

**Decoding**
Same as BPE: The text strings corresponding to each token ID are simply looked up and joined together.

## B   PSEUDOCODE FOR STOCHASTOK

---

**Algorithm 1** STOCHASTOK: Construction of `splits`

---

 1: **Require:** Tokenizer (e.g. `tiktoken`'s GPT-2 tokenizer)
 2: $\mathcal{V} \leftarrow$ Tokenizer vocabulary
 3: `splits` $\leftarrow \{\}$                                                              Initialize an empty dictionary
 4: **for each** token $s$ in $\mathcal{V}$ **do**
 5:     $t \leftarrow \text{encode}(s)$                                                                Get the token id
 6:     `splits`$[t] \leftarrow [\,]$                                                    Initialize empty list for this token
 7:     **for each** possible split index $i$ from 1 to $\text{len}(s) - 1$ **do**
 8:         $s_1, s_2 \leftarrow s[:i], s[i:]$                                          Split string $s$ into two substrings
 9:         **if** $s_1$ and $s_2$ in $\mathcal{V}$ **then**
10:             $t_1, t_2 \leftarrow \text{encode}(s_1), \text{encode}(s_1)$                        If both substrings are in the vocab
11:             `splits`$[t]$.append$((t_1, t_2))$                                         Add this possible split
12:         **end if**
13:     **end for**
14: **end for**

---

---

[7]WordPiece (Schuster & Nakajima, 2012) can be seen as a variant of BPE with merges during encoding chosen by token length rather than the original merge rules.

---

**Algorithm 2** STOCHASTOK: Tokenization

---

1: **Require:** Tokenizer
2: **Require:** `text`: The input text to tokenize
3: **Require:** `splits`: Dictionary of possible splits for each token
4: **Require:** `expand_prop`: Expansion proportion (e.g. $= 0.01$)
5: `tokenized` $\leftarrow$ Tokenizer(`text`)                                          Apply standard tokenization
6: `num_to_expand` $\leftarrow$ len(`tokenized`) $*$ `expand_prop`
7: **for** _ in $1 \cdots$ `num_to_expand` **do**
8:    $i \leftarrow$ randomInteger$(1, \text{len}(\texttt{tokenized}))$                       Choose a random position
9:    $t \leftarrow \texttt{tokenized}[i]$
10:    **if** $t$ in `splits` **and** `splits`$[t]$ not empty **then**
11:       $(t_1, t_2) \leftarrow$ randomChoice(`splits`$[t]$)                  Replace with a random split
12:       `tokenized` $\leftarrow$ `tokenized`$[1 : i - 1] + [t_1, t_2] + $ `tokenized`$[i + 1 :]$
13:    **end if**
14: **end for**
15: **return:** `tokenized`

---

## C   STOCHASTIC TOKENIZATION ILLUSTRATIVE EXAMPLE

Example vocabulary of base tokenizer:
```
vocabulary = [_, h, u, g, b, m, hu, ug, hug, bug]
```

Build `token_splits` which, for each token, contains a list of all possible pairs of component tokens that are themselves in the vocabulary.
```
token_splits = {
        ug:[(u,g)],
        hu:[(h,u)],
        hug:[(h,ug),(hu,g)],
        bug:[(b,ug)],
        ugs:[(ug,s)]
    }
```
Examples of possible expansions:
```
original:  [hug] → all possible expansions:  [hu g], [h ug], [h u g]
original:  [bug] → all possible expansions:  [b ug], [b u g]
original:  [m ug] → all possible expansions:  [m u g]
```

## D   LANGUAGE GAME QUESTIONS

```
Which word has the most letter 'n's?  These are the available options:
[ reason, step, continent, their].  Answer:  continent.

Which choice contains 'ec'?  The possible choices are:  [ was, children,
require, check].  Answer:  check.

Which option string starts with 'mo'?  The options:  [ case, ask, month,
event].  Answer:  month.

Which of the option words ends with 'ad'?  The option words are:  [ cost,
lead, south, sun].  Answer:  lead.

Which of the available choices is the longest?  These are the available
choices:  [ wild, dear, had, section].  Answer:  section.

Which string is the shortest?  The possible option words:  [ thought,
job, circle, nothing].  Answer:  job.
```

