# OpenReview forum: "StochasTok: Improving Fine-Grained Subword Understanding in LLMs"
_ICLR.cc/2025/Workshop/BuildingTrust — BuildingTrust_

### Official Review · Reviewer_cNaf · 2025-03-02
**Interesting application and patching a 'hole' in current LLM operations, simple but effective methodology**

**Rating:** 8
**Confidence:** 3

**Review:**

# Evaluation of the Work

## Summary of the Review
Overall, I find this a very interesting workshop paper. It attempts to patch a 'hole' in LLM performance that, while not substantial, can significantly influence the trust that users have in the models.

## Quality
**Pros:**
-  Overall, it is a solid argument
- Figures and descriptions of how the methods live up to current benchmarks are clear and concise

**Cons:**
-  I think section 2 was a little out of date in terms of the research that is currently going on. The most recent paper was 2020 and ignores a lot of the more recent explanations and evolutions in how tokenization is being considered in LLMs. For example, Ali et al. (2024) https://aclanthology.org/2024.findings-naacl.247/?link_id=7e4dc829-9b3e-43e3-9ed6-15767f1556be

## Clarity
**Pros:**
-  The right mix of information and 'extra' information in the appendix. I thought it was interesting and clear, but I am also glad the appendix was there. This is particularly relevant for section 3
- Figure 5 really made the point evident to me

**Cons:**
-  In lines 126/127, the authors refer to their "first experiment," but I see no evidence of further experiments. That's fine if it is only one, particularly for a workshop, but I just can not seem to identify what the series of experiments is referring to.

## Originality
**Pros**
-  Overall, tokenization is becoming an important sub topic in LLMs and in how to use this feature to improve performance
- This work makes a concrete, unique contribution in looking at a specific implementation within the field

**Cons:**
-  Seems to play off a lot of previous tokenization work. One such work was Singh & Strouse (2024) in https://arxiv.org/abs/2402.14903 that tackled tokenization for arithmetic expressions.
- Small question on how novel this is, as other similar cases in tokenization have been explored, however I have not seen this particular case discussed yet

## Significance & Relevance
**Pros:**
-  It highlighted how, if such simple tasks are failing, we are not likely to view an LLM as trustworthy, making it very relevant to the workshop
- It is an interesting point and significant in the sense that it provides a way to fix the issue for spelling situations that do not impact other evaluation metrics (an important & significant feature).

**Cons:**
-  It, on the surface, seems like a very small solution to a few edge cases (e.g. users are not likely to ask 'how many 'r's are in strawberries?).

---

### Official Review · Reviewer_UE7x · 2025-03-02
**StochasTok: Improving Fine-Grained Subword Understanding in LLMs**

**Rating:** 8
**Confidence:** 4

**Review:**

Very clever intervention to solve a particular type of problem that perplexes LLMs. Methodology reads robustly and the experiments seem thorough and well-designed. Overall good quality, clarify, and originality.

Some areas of improvement: It would be good to test this methodology on a larger model (maybe even GPT-2), although it is understandable that the paper did not do so due to the cost. Additionally, the analysis of the internal representations of words could be more clear—what exactly is being compared?

---

### Official Review · Reviewer_kzAR · 2025-03-02
**The approach is promising and outlined nicely.**

**Rating:** 7
**Confidence:** 3

**Review:**

### Summary

This paper provides a method for improving subword understanding of LLMs, based on the approach of stochastic tokenization. The method is computationally inexpensive, and provides said improvements.

### Strength

1. Paper is nicely written, and the overall flow of discussion is good.
2. The proposed method achieves significant gains in subword understanding task compared to standard training.
3. The computation cost of the method is minimal.
4. The performance on original benchmarks is not hindered, illustrating that stochastic tokenization is a promising approach.


### Weaknesses

1. Code is not provided. There is no way to reproduce the results.

---

### Decision · Program_Chairs · 2025-03-02

Accept